# Surface Functionalization of Non-Woven Fabrics Using a Novel Silica-Resin Coating Technology: Antiviral Treatment of Non-Woven Fabric Filters in Surgical Masks

**DOI:** 10.3390/ijerph19063639

**Published:** 2022-03-18

**Authors:** Chiaki Tsutsumi-Arai, Yoko Iwamiya, Reiko Hoshino, Chika Terada-Ito, Shunsuke Sejima, Kazuhiro Akutsu-Suyama, Mitsuhiro Shibayama, Zenji Hiroi, Reiko Tokuyama-Toda, Ryugo Iwamiya, Kouhei Ijichi, Toshie Chiba, Kazuhito Satomura

**Affiliations:** 1Department of Oral Medicine and Stomatology, Tsurumi University School of Dental Medicine, 2-1-3 Tsurumi, Tsurumi-ku, Kanagawa, Yokohama 230-8501, Japan; tsutsumi-c@tsurumi-u.ac.jp (C.T.-A.); terada-chika@tsurumi-u.ac.jp (C.T.-I.); tokuyama-r@tsurumi-u.ac.jp (R.T.-T.); nitou2284@icloud.com (K.I.); 2Choetsu Kaken Co., Ltd., 1-1-40 Suehiro-cho, Tsurumi-ku, Kanagawa, Yokohama 230-0045, Japan; y.iwamiya@choetsu.co.jp (Y.I.); ryugo@yk.rim.or.jp (R.I.); 3Department of Cultural Properties, Tsurumi University Faculty of Literature, 2-1-3 Tsurumi, Tsurumi-ku, Kanagawa, Yokohama 230-8501, Japan; hoshino-reiko@tsurumi-u.ac.jp; 4Biomedical Science Association, 2-20-8 Kamiosaki, Shinagawa-ku, Tokyo 141-0021, Japan; sejima@npo-bmsa.org; 5Neutron Science and Technology Center, Comprehensive Research Organization for Science and Society (CROSS), 162-1 Shirakata, Tokai-mura, Naka-gun, Ibaraki 319-1106, Japan; k_akutsu@cross.or.jp (K.A.-S.); m_shibayama@cross.or.jp (M.S.); 6Institute for Solid State Physics, The University of Tokyo, 5-1-5 Kashiwanoha, Kashiwa, Chiba 277-8581, Japan; hiroi@issp.u-tokyo.ac.jp; 7Research Center of Electron Microscopy, Tsurumi University School of Dental Medicine, 2-1-3 Tsurumi, Tsurumi-ku, Kanagawa, Yokohama 230-8501, Japan; chiba-t@tsurumiu.ac.jc

**Keywords:** COVID-19, SARS-CoV-2, mask, non-woven fabric, antiviral activity, silica-resin coating technology

## Abstract

Masks are effective for preventing the spread of COVID-19 and other respiratory infections. If antimicrobial properties can be applied to the non-woven fabric filters in masks, they can become a more effective countermeasure against human-to-human and environmental infections. We investigated the possibilities of carrying antimicrobial agents on the fiber surfaces of non-woven fabric filters by applying silica-resin coating technology, which can form silica-resin layers on such fabrics at normal temperature and pressure. Scanning electron microscopy and electron probe microanalysis showed that a silica-resin layer was formed on the fiber surface of non-woven fabric filters. Bioassays for coronavirus and quantitative reverse transcription-polymerase chain reactions (RT-PCR) revealed that all antimicrobial agents tested loaded successfully onto non-woven fabric filters without losing their inactivation effects against the human coronavirus (inhibition efficacy: >99.999%). These results indicate that this technology could be used to load a functional substance onto a non-woven fabric filter by vitrifying its surface. Silica-resin coating technology also has the potential of becoming an important breakthrough not only in the prevention of infection but also in various fields, such as prevention of building aging, protection of various cultural properties, the realization of a plastic-free society, and prevention of environmental pollution.

## 1. Introduction

A new coronavirus disease (COVID-19) caused by severe acute respiratory coronavirus 2 (SARS-CoV-2) was first reported in Wuhan, China, in December 2019 [1,2]. This virus was confirmed to transmit quickly from one human to another by multiple means, such as through droplets, aerosols, and even environmentally contaminated fomites [3]. On 11 March 2020, the World Health Organization (WHO) declared COVID-19 as a pandemic after 118,000 cases had been reported and more than 4000 people had died worldwide; the virus was confirmed in every continent except Antarctica. Currently, a total of five variants of SARS-CoV-2 have been identified, and the latest Omicron strain is more transmissible than any other, posing a major threat worldwide. There have been 273.39 million confirmed cases and 5.34 million deaths worldwide as of 21 December 2021 [4]. On 8 August 2020, Britain became the first country to receive a COVID-19 vaccine. Various types of COVID-19 vaccines are currently being administered around the world. However, the vaccination rates vary significantly across developed and developing countries, resulting in new regional differences and gaps.

Since SARS-CoV-2 is spread to vulnerable hosts through different means, such as droplets, aerosols, and through contact with contaminated surfaces [5,6,7,8,9], proper use of personal protective equipment (PPE) is essential for reducing the risk of infection in healthcare and community settings [10,11]. Given that the major pathways for infection come from droplets that transmit through the sneezing and coughing of symptomatic patients, and/or through conversation and speech near asymptomatic infected persons, wearing respiratory PPE such as masks is crucial for the effective prevention of SARS-CoV-2 infection [12]. Further, recent studies have shown that epithelial cells of the oral mucosa and salivary glands express angiotensin-converting enzyme-2 (ACE-2), a SARS-CoV-2 receptor [13,14] with saliva, which contains a significant number of SARS-CoV-2 as well as exudate from respiratory tracts [15,16]. These facts highlight the importance of appropriate use of respiratory PPE in preventing the spread of SARS-CoV-2 through communities. Unfortunately, there is currently no straightforward and conclusive evidence for the relative efficacy of the various types of respiratory PPE, such as N95 respirators, surgical, and cloth masks to reduce the risk of SARS-CoV-2 infection, as a result of identified mask types that have been investigated [17]. However, the more plentiful data on SARS-CoV-1/MERS-CoV infection indicate that consistent use of masks reduces the risk of SARS-CoV-1/MERS-CoV infection [18,19]. Other studies have reported that masks may minimize the risk of clinical diseases such as influenza [20]. Thus, developing respiratory PPE with adequate antiviral activity to prevent transmission of SARS-CoV-2 and other infectious pathogens that cause respiratory disease is crucial and valuable. Currently, several methods, such as spraying with antibacterial agents or with silver ion solution, or applying photocatalysts, have been developed to make the surfaces of masks antibacterial. Each method has problems, such as short duration of effect, or complexity of processing and production [21]. Therefore, it is thought that applications of technology which more easily enable non-woven fabric filters of surgical masks to become antimicrobial will result in the utilization and popularization of antibacterial/antiviral masks.

The most common inorganic material on earth is silicon dioxide (SiO_2_), also called silica. It is employed as a coating to protect various reactive substances from the environment, since it is chemically stable and harmless to organisms [22,23]. The silica coating is a glass thin layer consisting of Si-O-Si siloxane bonds that peels easily from substrate surfaces, since it is hard and brittle. However, we recently reported a new silica-resin coating technology [24]. The film produced by this technology is not a hard silica glass with four siloxane bonds per Si molecule, but a sparse three-dimensional glass network structure comprised of siloxane bonds. A flexible and stable glass thin film can be produced on the surface of cellulose fibers by forming covalent bonds with existing hydroxyl groups in cellulose and other substances. The presence of a significant number of alkyl groups, such as methyl groups derived from methyltrimethoxysilane (MTMS), makes it possible to form fine spaces within the glass network and thus convey various substances through it.

In this study, by utilizing the unique properties of the thin glass layer formed by this new silica-resin coating technology, i.e., stability, flexibility, and microspace-forming ability on materials, we attempted to vitrify the surface of the non-woven fabric filter of surgical masks. We then loaded the material with various antimicrobial reagents, such as cetylpyridinium chloride (CPC), chlorhexidine gluconate, benzalkonium chloride, povidone-iodine, hinokitiol, and grapefruit seed extract, to explore the possibility of developing a high-performance non-woven fabric filter with high antiviral properties under more convenient and simple conditions than ever. 

## 2. Materials and Methods

### 2.1. Non-Woven Fabric Filter in Surgical Masks

In the following experiments, non-woven fabric filters were used, which are fabricated with polypropylene in the middle layer of medical surgical masks (ASKUL Corporation, Tokyo, Japan) with a three-layer structure.

### 2.2. Processing of Non-Woven Fabric Filters Using Silica-Resin Coating Technology

The stock solution used in this study contained 27.0% (*w*/*w*) methyltrimethoxysilane (MTMS: CH_3_Si (OCH_3_)_3_) oligomer (Momentive. Inc., New York, NY, USA), 17.0% (*w*/*w*) tetramethoxysilane oligomer (TMOS: (CH_3_O)_4_Si, Mitsubishi Chemical Corp., Tokyo, Japan), 6.0% (*w*/*w*) tetraisopropyl titanate (TPT: [(CH_3_)_2_CHO]_4_Ti), Nippon Soda Co., Ltd., Tokyo, Japan), 0.001% (*w*/*w*) platinum catalyst (CM670, Momentive. Inc., New York, NY, USA), 1.399% (*w*/*w*) acetic acid (Showa Denko, Tokyo, Japan), and 48.6% (*w*/*w*) isopropanol (IPA, Sankyo Chemical Industry, Co., Ltd., Osaka, Japan). The basal solution was prepared by diluting 27.5 g stock with 72.5 g of IPA. Each processing solution was prepared by adding either hinokitiol (HKL), or an antimicrobial agent of cetylpyridinium chloride (CPC, FUJIFILM Wako Chemical Corp., Osaka, Japan), grapefruit seed extract (grapefruit seed extract (GSE), Will Tool, Biochemical Technical Laboratory Co., Ltd., Niigata, Japan), chlorhexidine gluconate (CHX, FUJIFILM Wako Chemical Corp., Osaka, Japan), benzalkonium chloride (BZC, FUJIFILM Wako Chemical Corp., Osaka, Japan), or povidone-iodine (PI, Kenei Pharmaceutical Co., Ltd., Osaka Japan), to the basal solution. Each non-woven fabric filter (15 cm × 15 cm, 0.56 g, ~0.10-millimeters thick) received about 5 g of each processing solution dropwise coated with a bar coater and dried at room temperature for 30 min. The thickness and weight of each treated filter were measured (Table 1) after treatment. Some filters were treated with the basal solution without any antimicrobial agents as a control. Brown changes in the filter color were observed after povidone-iodine (PI) treatment, but no changes in filter color were observed with other antimicrobial agents.

### 2.3. Scanning Electron Microscopy

To avoid the excess part of the coating agent, the specimens for observation were chosen at random from the inner part, which is more than 1 cm away from the edge. Each specimen was placed on an aluminum stub, sputtered coated with gold in an ion sputter coater (IB-3, Eiko Engineering, Tokyo, Japan), and then examined using a scanning electron microscope (SM -300, TOPCON Corp., Tokyo, Japan).

### 2.4. Electron Probe Macroanalysis

Electron probe macroanalysis was performed for O, C, Ti, and Si using an electron probe micro-analyzer (EPMA JEM8900, JEOL) at an acceleration voltage of 15.0 kV and an irradiation current of 2.5e-8 mA to measure the distribution state of elements.

### 2.5. Air Permeability Test of Silica-Resin Coating Technology-Treated Non-Woven Fabric Filters

The air permeability of the entire mask (i.e., outer layer + middle layer + inner layer) was measured using FX 1096 LabAir (Textest AG, Schwerzenbach, Switzerland) based on the JIS L1096A protocol (Flagyl method), after silica-resin coating technology processing of the non-woven fabric filters of the middle layer of the three-layered surgical masks.

### 2.6. Antiviral Efficacy of Non-Woven Fabric Filters Treated with Silica-Resin Coating Technology and Antimicrobial Agents

Each treated filter’s antiviral activity was evaluated under a modified version of ISO 18184:2019. The treated and control filters were inoculated with 200 μL of human coronavirus 229E solution (ATCC VR-740, 1 × 10^7^ TCID_50_/mL) and maintained at 25 °C for 2 h. The virus attached to each filter was flushed into 20 mL of phosphate-buffered saline (PBS) by vortexing, and a quantitative reverse transcription-polymerase chain reaction (RT-PCR) was performed on an aliquot of each solution.

### 2.7. Reverse Transcription-Polymerase Chain Reaction

One-step RT-PCR was performed using a PCR1100 device. The procedure of the PCR1100 device used was based on previously reported procedures [25]. In a 20-microliter reaction solution containing 1× Reaction Buffer, 0.25 or 0.50 µL of RT Enzyme Mix, 1.0 µL DNA Polymerase (THUNDERBIRD™ Probe One-step qRT-PCR Kit, TOYOBO Co. Ltd., Osaka, Japan), and a final concentration of primer/probe for each target, 3 µL of a sample (PBS containing human coronavirus 229E washed out from treated non-woven fabric filters) was amplified (Table 1). The qRT-PCR was performed under the following conditions: at 50 °C in 150 s and 95 °C in 15 s, RT incubation and enzyme activation, respectively. Afterward, it was cycled fifty times at 95 °C in 3.5 s for denaturation, and at 60 °C in 8 s for annealing and amplification. Each filter’s TCID_50_ was calculated based on its corresponding Ct value.

## 3. Results

### 3.1. Microstructure of Silica-Resin Coating Technology-Treated Non-Woven Fabric Filters

The thicknesses and weights of 15 cm × 15 cm non-woven fabric filters processed by silica-resin coating technology were 0.12 to 0.22 mm and 0.90 to 1.06 g, respectively (Table 1). Scanning electron microscopy was used to observe the morphology of the non-woven fabric filters treated with each working solution. Polypropylene fibers with various diameters ranging from 1 to 8 μm were intricately intertwined (Figure 1G) in the original non-woven fabric filter structure. Comparing silica-resin coating technology with the untreated control, a slight enlargement in diameter was observed of non-woven fabric filter fibers treated with CPC (Figure 1A), GSE (Figure 1B), CHX (Figure 1C), BZC (Figure 1D), PI (Figure 1E), and HKL (Figure 1F). Further, the formation of amorphous structures was also observed among fibers in certain areas of non-woven fabric filters treated with CPC (Figure 1A), GSE (Figure 1B), CHX (Figure 1C), BZC (Figure 1D), and PI (Figure 1E). However, such structures were not observed in non-woven fabric filters treated with HKL (Figure 1F).

The original non-woven fabric filter showed a structure in which polypropylene fibers were intricately intertwined. There were noted slight enlargements in fiber diameters of non-woven fabric filters treated with antimicrobial agents of CPC (A), GSE (B), CHX (C), BZC (D), PI (E), and HKL (F) using silica-resin coating technology compared with untreated control. The formation of amorphous structures was also observed among fibers in some areas of non-woven fabric filters treated with CPC (A), GSE (B), CHX (C), BZC (D), and PI (E). In contrast, such structures were not seen in non-woven fabric filters treated with HKL (F).

### 3.2. Electron Probe Microanalysis of Silica-Resin Coating Technology-Treated Non-Woven Fabric Filters

Elemental mapping with an electron probe microanalyzer showed the presence of Si and O on the fibers of non-woven fabric filters, indicating that a thin layer of silica-resin formed on the surface of the non-woven fabric filter treated with silica-resin coating technology (Figure 2 and Figure 3).

### 3.3. Air Permeability Test of Silica-Resin Coating Technology-Treated Non-Woven Fabric Filters

Treatment using silica-resin coating technology with microbial agents such as CPC, GSE, CHX, BZC, and PI decreased the air permeability of masks by a similar extent. In contrast, treatment with a basal solution containing HKL did not decrease air permeability (Figure 4). These results were consistent with the fact that amorphous structures that induce clogging of filters were not observed in HKL-treated non-woven fabric filters.

The air permeability of masks decreased by a similar extent after treatment using silica-resin coating technology with microbial agents, i.e., CPC, GSE, CHX, BZC, and PI. In contrast, treatment with a basal solution with HKL did not cause a decrease in air permeability. These findings were consistent with the fact that amorphous structures which may cause clogging of filters were not observed in HKL-treated non-woven fabric filters (Figure 1F).

### 3.4. Antiviral Efficacy of Non-Woven Fabric Filters Treated with Silica-Resin Coating Technology and Antimicrobial Agents

Although the antiviral activities of surfaces of non-woven fabric filters were slightly different based on which microbial agents were employed, silica-resin coating technology could impart antiviral activity to the surface of all non-woven fabric filters treated with different antimicrobial agents. In Ct values, the number of human coronaviruses 229E adhering to non-woven fabric filter surfaces treated with CPC, GSE, CHX, BZC, and PI were >50, >50, 45.2, >50, and 40.4, respectively. Human coronavirus 229E adhering to non-woven fabric filter surfaces treated with CPC, GSE, CHX, BZC, and PI to MRC-5 cells had infectivity titers of <2.5 × 10^0^, <2.5 × 10^0^, 2.5 × 100, <2.5 × 10^0^, and 4.5 × 10^1^/mL in TCID_50_, respectively. The untreated non-woven fabric filter indicated a Ct value of 23.9 and a TCID_50_ of 7.5 × 10^5^/mL, which indicate that the untreated filter itself had no antiviral activity. Further, in the non-woven fabric treated only with silica-resin coating technology without any antimicrobial agents, the Ct and TCID_50_ values were 24.5 and 4.5 × 10^5^/mL, respectively. This indicates that the treatment with only basal solution such as silica resin itself forms on the surface of non-woven fabric filters without imparting any antiviral activity on the surface of non-woven fabric filters (Table 2).

Antiviral activity of treated and untreated filters was assessed in accordance with ISO 18184:2019. Each treated filter and control filter were inoculated with 200 μL of the solution of human coronavirus 229E, and the virus attached to each filter was retrieved into 20 mL of PBS. The virus amount (Ct value) was measured by qRT-PCR. The calculated TCID_50_ value of each filter was determined based on the corresponding Ct value. The inhibition ratio to control was also calculated for each filter. 

## 4. Discussion

The fiber surface of a non-woven fabric filter used for a surgical mask can be functionalized under simple conditions at ambient temperature and pressure using silica-resin coating technology. However, doing so significantly reduces the roughness of the filter as well as its air permeability, except for filters treated with HKL. Filter roughness directly affects filtration efficiency for masks used for infection control, such as bacterial filtration efficiency, particulate filtration efficiency, and viral filtration efficiency [26]. Thus, when applying this technique to the manufacturing process of masks with the performance requirements in a medical field, it is crucial to set the roughness of the non-woven fabric filter in advance of the manufacturing process in order for non-woven fabric filters to achieve appropriate filtration efficiency and air permeability of masks processed by silica-resin coating technology. However, since the surface of polypropylene non-woven fabric involves simple processing techniques, it is expected to significantly contribute not only to improving mask function, but also to enhance environmental infection countermeasures of various medical devices, clinical laboratory equipment, and medical facilities. The exact reason for why amorphous structures did not form in HKL-treated non-woven fabric filters is unknown, but one possibility is the physicochemical properties of HKL. It should be clarified in future studies how treatment with only HKL does not result in the formation of amorphous structures.

Results showed that silica-resin coating technology facilitated the loading of several antimicrobial agents onto the surface of polypropylene non-woven fibers, and that all the substances observed in this study retained their intrinsic antimicrobial effects. This was verified by the absence of antiviral activity in non-woven fabric filters treated with silica-resin coating technology’s basal solution without any antimicrobial agents. The antiviral activities imparted to non-woven fabric filters differed slightly among the antimicrobial agents used in the quantitative determination of virus quantity (Ct value) by qPCR and the infectivity titer (TCID_50_) of the virus to cultured cells. However, since the antimicrobial agents examined in this study had different molar concentrations, extensive discussions regarding their differences in antiviral activity will be avoided and addressed in more specialized areas of infection control in the future.

The process by which silica-resin coating technology facilitates loading of various antimicrobial agents on the surface of polypropylene fibers is unclear in this study. However, a technique for forming silica-resin thin films on material surfaces using silica-resin coating technology was recently reported [24]. Iwamiya et al., reported that by the hydrolysis reaction with water molecules present on the material surface and in the surrounding environment, oligomers of alkoxysilane, such as methyltrimethoxysilane (MTMS; CH_3_Si (OCH_3_)_3_), formed a thin silica-resin film with strong Si-O-C bonds on the material surface. This was followed by condensation polymerization (dehydration reaction) between the hydroxyl groups of the oligomers and those generated on the material surface. Further, it was suggested that the presence of alkyl groups in the silica-resin film, which correspond to the methyl groups of MTMS, cause a cage-like spatial structure to form. Based on this report, there are two possible means by which silica-resin coating technology loads various antimicrobial agents onto the surface of polypropylene fibers. First, the antimicrobial agent’s hydroxyl groups participated in the hydrolysis reaction with TMTS to produce a compound with the TMTS oligomer at the initial stage of the reaction, which was then integrated into the silica-resin on the material surface through a series of reactions. Among the five antimicrobial agents used in this study, GSE, CHX, and hinokitiol have hydroxyl groups in their chemical structures. The second possible loading mechanism for polypropylene fibers involves antimicrobial molecules being integrated into the cage-like spatial structure of the silica-resin thin film formed on the material surface. Although this study has not clarified which of these two possibilities is the true mechanism, since various antimicrobial agents were loaded irrespective of the presence or absence of hydroxyl groups, the second mechanism appears to play a significant role. Analysis employing FT-IR, ^1^H-NMR, X-ray reflectivity, neutron reflectivity, and other techniques would be valuable in elucidating this process. Further research will be conducted in the future.

## 5. Conclusions

This study demonstrated the antiviral functionalization of non-woven fabric filters used in surgical masks and suggested the great potential role of silica-resin coating technology in the control of environmental infections. Specifically, using silica-resin coating technology has shown that (1) a silica-resin thin film can be formed on the surface of a non-woven fabric filter in a short time under simple environmental conditions of air temperature and pressure, and (2) various substances having antimicrobial effects can be easily loaded onto non-woven fabric filter surfaces while maintaining their functionality. Based on these results, it is expected that by making the best use of this technology, we will be able to efficiently, effectively, and economically load antiviral effects onto non-woven fabric filters used in surgical masks and other products, and contribute to curbing the spread of COVID-19 infection, which is currently a critical global issue. Moreover, judging from the fact that it was possible to vitrify the surfaces of not only cellulose but also of polypropylene, which is an organic polymer, to carry antibacterial agents, it has become clear that various functions, such as rust prevention, light resistance, and weather resistance may also be easily imparted to the surfaces of various materials according to their purpose. This strongly suggests the possibility that this technology can be an important breakthrough not only in the prevention of infection, but also in various fields such as the prevention of building deterioration, the protection of various cultural properties (paper, wood, and stone cultural properties), the realization of a plastic-free society, and the prevention of environmental pollution. 

## Figures and Tables

**Figure 1 ijerph-19-03639-f001:**
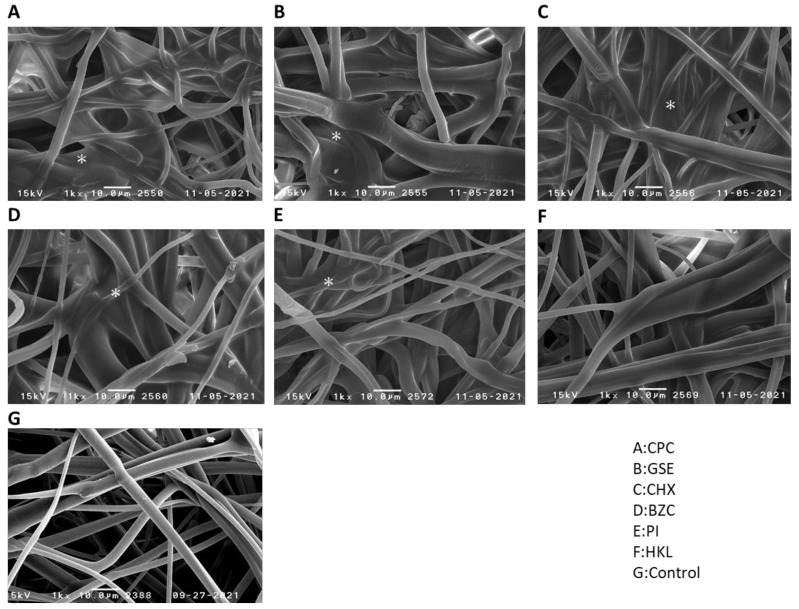
Scanning electron microscopy (SEM) images of the fiber surfaces of non-woven fabric filters treated with CPC (**A**), GSE (**B**), CHX (**C**), BZC (**D**), PI (**E**), or HKL (**F**). Amorphous structures (*) were observed among fibers in certain areas of non-woven fabric filters treated with CPC (**A**), GSE (**B**), CHX (**C**), BZC (**D**), and PI (**E**), but not with HKL (**F**) and (**G**) control.

**Figure 2 ijerph-19-03639-f002:**
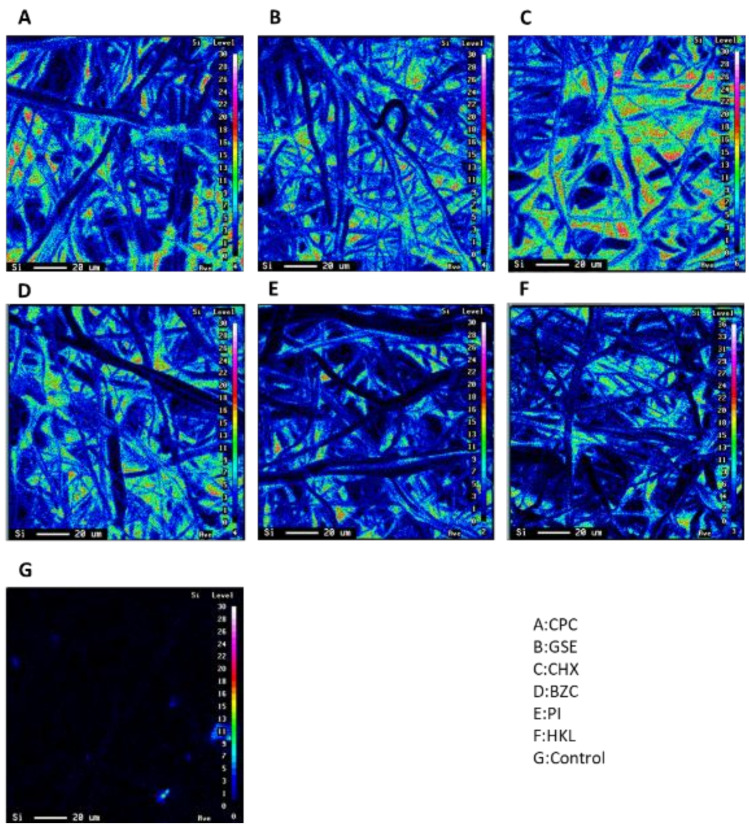
Elemental mapping of the fiber surfaces of non-woven fabric filters treated with CPC (**A**), GSE (**B**), CHX (**C**), BZC (**D**), PI (**E**), or HKL (**F**). The presence of Si on the fibers of non-woven fabric filters was confirmed in (**A**–**F**), but not in (**G**) (Control).

**Figure 3 ijerph-19-03639-f003:**
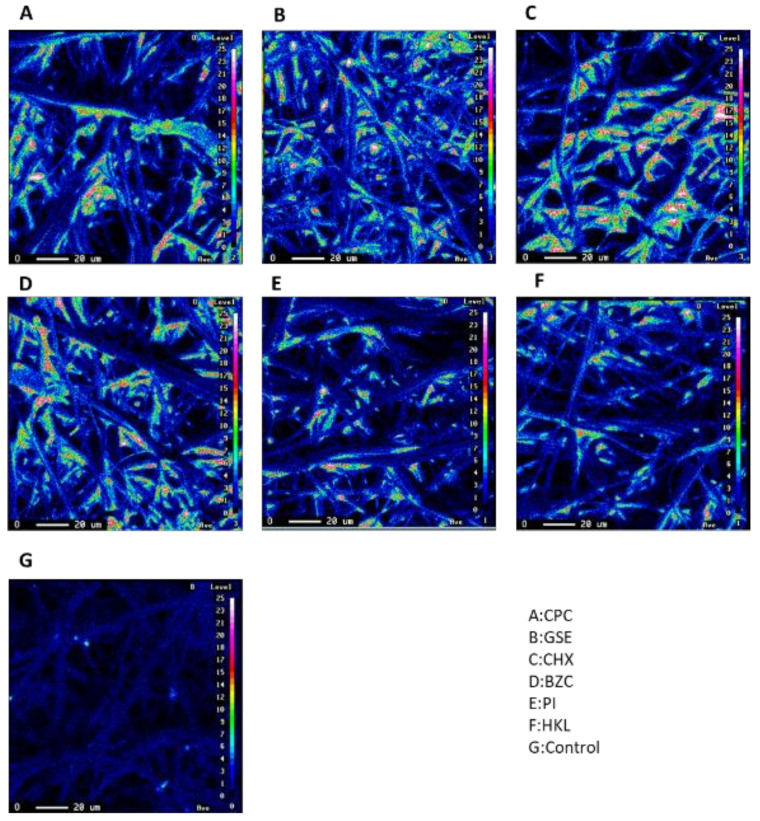
Elemental mapping of the fiber surfaces of non-woven fabric filters treated with CPC (**A**), GSE (**B**), CHX (**C**), BZC (**D**), PI (**E**), or HKL (**F**). The presence of O on the fibers of non-woven fabric filters was confirmed in (**A**–**F**), but not in (**G**) (Control).

**Figure 4 ijerph-19-03639-f004:**
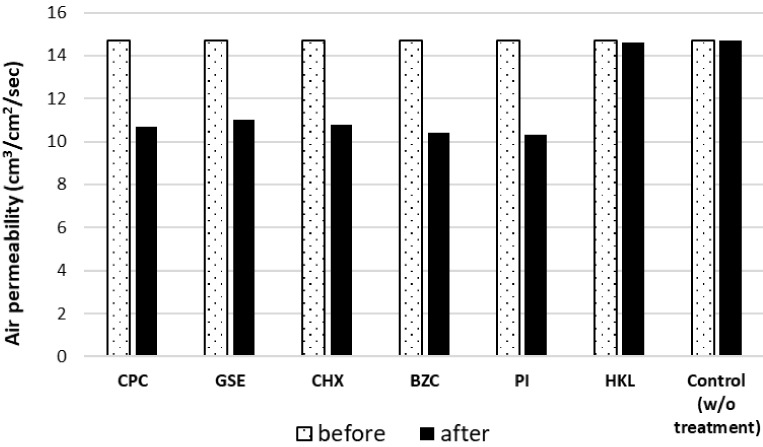
Air permeability of masks before/after silica-resin coating technology treatment.

**Table 1 ijerph-19-03639-t001:** Antimicrobial agents for processing of non-woven fabric filter.

Antimicrobial Agents	Amount (g)	pH	Thickness (mm)	Weight (g)
Cetylpyridinium chloride	2.0	6	0.19	0.97
Grapefruit seed extract	2.0	6	0.18	0.99
Chlorhexidine gluconate (20%)	3.0	6	0.15	1.06
Benzalkonium chloride (10%)	16.0	6	0.12	1.01
Povidone iodine (10%)	10.0	5.5	0.18	0.90
Hinokitiol (2%)	25.0	6	0.22	0.90

To prepare each working solution, 100 g of basal solution was added to the indicated amount of antimicrobial agent, and a non-woven fabric filter (15 cm × 15 cm, ~0.10 mm in thickness, 0.56 g) was dipped into each working solution. pH: each operating solution’s pH value; thickness: post-processing thickness; weight: post-processing weight.

**Table 2 ijerph-19-03639-t002:** Antiviral efficacy of non-woven fabric filters treated with silica-resin coating technology and antimicrobial agents.

Antimicrobial Agent	Amount (g)	Ct Value	Calculated TCID_50_ (/mL)	Inhibition Efficacy (%)
Cetylpyridinium chloride (CPC)	2.0	ND	<2.5 × 100	>99.9994
Grapefruit seed extract (GSE)	2.0	ND	<2.5 × 100	>99.9994
20% Chlorhexidine gluconate (CHX)	3.0	45.2	2.5 × 100	99.9994
10% Benzalkonium chloride (BZC)	16.0	ND	<2.5 × 100	>99.9994
10% Povidone iodine (PI)	10.0	40.4	4.5 × 101	99.9998
2% Hinokitiol (HKL)	25.0	47.8	<2.5 × 100	>99.9994
Control (non-woven fabric filter w/o any treatment)	N/A	23.9	7.5 × 105	N/A
Control (non-woven fabric filter treated by silica-resin coating technology without any antimicrobial agent)	N/A	24.5	<4.5 × 105	N/A

## Data Availability

The data presented in this study are available on request from the corresponding author.

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
