# Peer review of "Surface Functionalization of Non-Woven Fabrics Using a Novel Silica-Resin Coating Technology: Antiviral Treatment of Non-Woven Fabric Filters in Surgical Masks"

_ijerph, 2022, doi:10.3390/ijerph19063639_

Round 1
Reviewer 1 Report
- Please re-write the abstract focusing on what is new and what is the achievement. The abstract should be informative by presenting both quantitative and qualitative information. Some quantitative information should be added.
- Obtained results must be compared with other related literature. Please discuss the advantages and disadvantages of the different materials and highlight your main achievement.
- The introduction section must be comprehensive. The author should re-word this section to improve the readability and continuity and clarify what is meant. The necessity of research should be described clearly. Furthermore, some relevant papers should be cited and the lack of literature in the area of research should be mentioned.
- I suggest adding the schematic representation for the experimental procedure.
- Figure 1, SEM images did not show significant differences. How could you confirm the successful coating of silica resin? I suggest providing EDX spectra and maps.
- Figure 2 is wrong. It should be plotted by a bar chart, not a line.
- More investigations are needed.
Reviewer 2 Report
The present manuscript entitled “Surface Functionalization of Nonwoven Fabrics Using a Novel Silica-Resin-Coating Technology: Antiviral Treatment of Nonwoven Fabric Filters in Surgical Masks” Chiaki Tsutsumi-Arai et al., describes the possibility of carrying some antimicrobial agents on the fiber surface of a nonwoven fabric filter used for a surgical mask by applying silica-resin coating technology, which can form a silica-resin layer on the surface of various materials at normal temperature and pressure. Furthermore, the results indicate that by processing the nonwoven fabric by silica-resin coating technology, it is feasible to verify the surface and transport the functional substance on it. Therefore, I recommend it for publication. However, certain Minor issues are detailed below which need to be addressed before its final acceptance in the International Journal of Environmental Research and Public Health.
Comment 1: There are so many typographical errors in the manuscript text, so authors need to correct them in the revised manuscript.
Comment 2: The abstract is poorly written, should be edited. It must summarize well the obtained results.
Comment 3: In the introduction section add some more recent literature to strengthen the section.
Comment 4: In the whole manuscript the authors must be taken care of the superscripts and subscripts and abbreviations, for e.g. Page 7, Line 259, CH3Si(OCH3)3 etc.,
Comment 5: Section 3.1: SEM results explanation should be discussed wider and compared with the other studies.
Comment 6: The conclusions section is too short, the authors should revise it.
Comment 7: Reference style is not uniformly maintained, there are so many superscripts and subscript errors, so correct it and maintain the uniformity.
Reviewer 3 Report
This manuscript proposed an interesting method to prepare antiviral coating to the fibers of the face mask. The experiments were properly designed. However, the writing of the manuscript can be improved. I would like to review it again after the authors addressing the following comments. Otherwise, I could not suggest the accept of this submission.
1. In the introduction, the author should explain why the flexible silica-resin is important for incorporating antimicrobial reagents onto the mask. For example, cannot the antimicrobials be directly coated onto the fibers or what methods or resins have been used by other researchers, and what is the potential advantage of this silica-resin.
2. page 5, could the author indicate in the SEM pictures the amorphous structure? I couldn't a big difference between the sample treated with HKL with other samples. Could the authors also provide some explanation of how the amorphous is formed and why it didn't form for the HKL?
3. page 6 one paragraph is repeated with one paragraph on page 5. I suggest the authors carefully re-read the manuscript and prevent any such mistakes.
4. page 6 line 208. the term "CT value" should be Ct for consistency.
Round 2
Reviewer 1 Report
The manuscript can be published, as the reviewer's concerns have been addressed.
Reviewer 3 Report
Thanks for the revision. It can be accepted.